# Mitigating Social Hazards: Early Detection of Fake News via Diffusion-Guided Propagation Path Generation

## ABSTRACT

The detection of fake news has emerged as a pressing issue in the era of online social media. To detect meticulously fabricated fake news, propagation paths are introduced to provide nuanced social context to complement the pure semantics within news content. However, existing propagation-enhanced models face a dilemma between detection efficacy and social hazard. In this paper, we investigate the novel problem of early fake news detection via propagation path generation, capable of enjoying the merits of rich social context within propagation paths while alleviating potential social hazards. In contrast to previous discriminative detection models, we further propose a novel generative model, *DGA-Fake*, by simulating realistic propagation paths based on news content before actual spreading. A guided diffusion module is integrated into *DGA-Fake* to generate simulated user interaction sequences, guided by historical interactions and news content. Evaluation across three datasets demonstrates the superiority of our proposal. Our code is available in *https://anonymous.4open.science/r/DGA-Fake-1D5F/*.

## CCS CONCEPTS

• **Information systems → Multimedia information systems**.

## KEYWORDS

Fake News Detection, Propagation Path Generation, Diffusion

## 1 INTRODUCTION

Fake news detection has garnered significant attention in recent years, driven by the rapid development of online social media platforms [14]. The widespread and rapid dissemination of toxic fake news poses a potentially immeasurable social hazard to both users and society. Traditional fake news detection methods [4, 18, 22, 26] generally focus on capturing the semantics within the news content via deep modeling architectures as illustrated in Figure 1 (a). In order to evade detection effectively [29], recent fake news is often meticulously fabricated by skilled malicious publishers, resulting in their indistinguishability for detection models that solely rely on semantic patterns [11].

In efforts to bolster detection capabilities, the propagation path is introduced as a complement to pure semantics [34, 41]. The propagation path encompasses user interactions during the dissemination of news on social networks [18], reflecting temporal and structural

*ACM MM, 2024, Melbourne, Australia*
© 2024 Copyright held by the owner/author(s). Publication rights licensed to ACM.
ACM ISBN 978-x-xxxx-xxxx-x/YY/MM
https://doi.org/10.1145/nnnnnnn.nnnnnnn

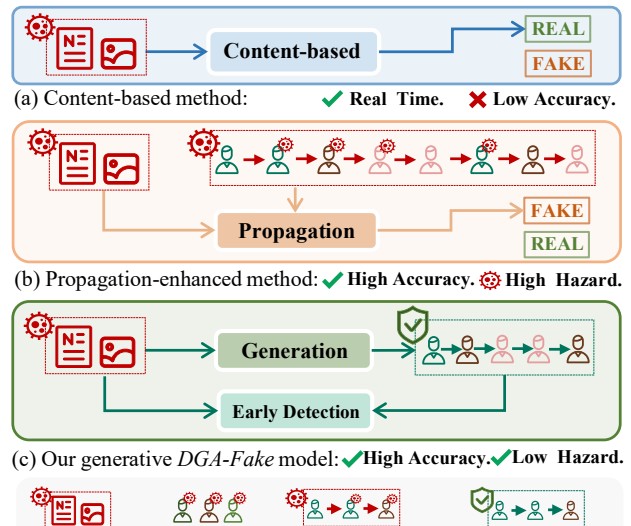

Figure 1: Comparison of different detection methods.

features of news propagation. Such propagation paths are capable of providing extra social context to facilitate the detection of fake news. For example, the propagation paths of fake news tend to be wider, deeper, and contain more social bots than real news [31], which is a significant clue to verify the fake news.

While the integration of propagation paths enhances the detection of well-fabricated news, such methods inherently face the dilemma between **detection efficacy** and **social hazard**. The abundant social context within extensive propagation paths is crucial for successful detection. On the other side of the coin, sufficient propagation paths also indicate that the fake news has already achieved widespread dissemination, leading to substantial social hazards, as depicted in Figure 1 (b). Detecting fake news that has already spread widely and caused significant social damage is somewhat futile. To mitigate this dilemma and identify fake news at an early stage, various forms of social context information have been introduced, such as user profiling [12, 32], publisher credibility assessment [42], and user comments analysis [24]. However, the acquisition of additional auxiliary information is time-consuming and labor-intensive, limiting its applicability in data-sparse scenarios. Moreover, the global structures and temporal features within the limited propagation paths remain insufficient.

The incorporation of propagation paths enhances detection performance, albeit accompanied by associated social hazard stemming from the spread of fake news. This brings us a question: *Can we leverage the rich social context within the propagation paths to enhance the early-stage fake news detection?* Different from existing discriminative models, our motivation lies in predicting and generating the forthcoming propagation paths based on the multimodal

content and propagation patterns gleaned from previous news. As illustrated in Figure 1 (c), our objective is to discern the patterns beneath the news propagation process, which enables the simulation of propagation paths, thereby facilitating the early detection of emerging news. This generation-based strategy does not necessitate additional social context and is anticipated to yield rich structural and temporal propagation features.

However, designing generative models for propagation path generation is non-trivial due to the two reasons. Firstly, it is challenging for generative models to accurately represent the distribution of discrete propagation paths. For example, the GAN-based models often encounter instability issues [3, 16], while the VAE-based models are susceptible to posterior collapse [35, 43]. Secondly, the diversity and complex connections inherent in user interactions within propagation paths pose a challenge for traditional generative models due to their limited capacity for diverse representation [40].

Based on such desiderata, we resort to denoising diffusion models [9] as solutions, which model discrete user interactions to generate diverse and high-quality propagation paths. However, directly applying vanilla denoising diffusion models to the studied task is undesirable. Firstly, the denoising process lacks controllable conditions. It is non-trivial to regulate the generation process to generate reasonable and informative propagation paths. Secondly, it is challenging to integrate multi-condition features and incorporate them into the generation process. Finally, modeling structural and temporal features from propagation paths simultaneously while enhancing detection performance poses significant challenges.

In this paper, we propose a novel _Diffusion Guided Propagation Augmentation Fake News Detection_ (DGA-Fake) model for multimodal fake news detection. Diverging from the prevailing discriminative detection methods, we propose simulating realistic propagation paths based on news content at the early detection stage. The simulation of user interaction paths is facilitated by a novel guided diffusion module. This module aids in stepwise denoising to produce reasonable next interaction user embeddings from randomly sampled Gaussian noise. Specifically, the denoising phase is guided by historical user interaction paths and news content, rendering it autoregressive in generating long propagation paths. Moreover, we propose a propagation path enhanced detection module, which focuses on the user sequence hypergraph to learn temporal depth information and the propagation directed graph to model propagation global structures. _DGA-Fake_ is extensively evaluated over three datasets, and the experimental results demonstrate its superiority.

Our contributions are summarized as follows:

- To the best of our knowledge, we are the first to investigate the novel problem of early fake news detection utilizing propagation path generation, enjoying the merits of rich social context while alleviating potential social hazards.
- We propose a novel _DGA-Fake_ model to generate reasonable propagation paths via controllable guided diffusion, and further integrate propagation global structure and temporal depth information through a propagation path enhanced fake news detection module.
- Extensive experiments on three real-world datasets reveal that our proposal consistently outperforms SOTA fake news detection baselines.

## 2 PRELIMINARY

### 2.1 Problem Definition

_Definition 2.1 (Fake News Detection)._ A news item is represented as $n = \{n^T, n^V\}$, with $n^T$ and $n^V$ denoting the textual and visual content, respectively. The set of users on the social platform is denoted as $U = \{u_1, u_2, \cdots, u_{|U|}\}$. The sequential path $S_i = \{n_0, u_1, u_2, \cdots, u_m\}$ denotes a single propagation sequence in chronological order, where $n_0$ serves as the starting point. All historical sequential paths of news $n$ are combined into the propagation path, represented as $\mathcal{P} = \{S_i\}$. The objective of fake news detection is to learn a probability distribution $\mathbb{P}(y|n, \mathcal{P})$, where $y = 0$ denotes fake news and 1 stands for real news.

_Definition 2.2 (Fake News Early Detection)._ At the early detection stage, the existing propagation paths are limited and may even be nonexistent. Specifically, the sequence $S_i = \{n_0\}$ signifies a lack of user interactions for the news item $n$. We aim to generate propagation paths and effectively discriminate fabricated news in the early stages, which is formalized as follows:

$$\mathbb{P}(y \mid n, \mathcal{P}) = \mathbb{P}(y \mid n, \mathcal{P}')\mathbb{P}(\mathcal{P}' \mid n, \mathcal{P}), \quad (1)$$

where $\mathcal{P}'$ is the simulated propagation paths base on news item $n$ and historical propagation path $\mathcal{P}$.

### 2.2 Diffusion Model

In this section, we will introduce the fundamental concept of the diffusion model [9].

2.2.1 _Noising Process._ The forward process of diffusion model is a Markov Chain that gradually adds Gaussian noise to $\mathbf{x^0}$:

$$q\left(\mathbf{x}^1, \cdots, \mathbf{x}^T \mid \mathbf{x}^0\right) = \prod_{t=1}^{T} q\left(\mathbf{x}^t \mid \mathbf{x}^{t-1}\right),$$
$$q\left(\mathbf{x}^t \mid \mathbf{x}^{t-1}\right) = \mathcal{N}\left(\mathbf{x}^t; \sqrt{1 - \beta_t}\mathbf{x}^{t-1}, \beta_t\mathbf{I}\right), \quad (2)$$

where $T$ is the number of steps to add noise and $[\beta_1, \beta_2, \ldots, \beta_T]$ are the variance schedule. It is difficult to learn $\beta_T$ directly, so a reparameterization trick [15] is adopted and transforms the forward process as follows:

$$q\left(\mathbf{x}^t \mid \mathbf{x}^0\right) = \mathcal{N}\left(\mathbf{x}^t; \sqrt{\bar{\alpha}_t}\mathbf{x}^0, (1 - \bar{\alpha}_t)\mathbf{I}\right),$$
$$\mathbf{x}^t = \sqrt{\bar{\alpha}_t}\mathbf{x}^0 + \sqrt{1 - \bar{\alpha}_t}\boldsymbol{\epsilon}, \quad (3)$$

where $\alpha_t = 1 - \beta_t$, $\bar{\alpha}_t = \prod_{s=1}^{t} \alpha_s$ and $\boldsymbol{\epsilon} \sim \mathcal{N}(\mathbf{0}, \mathbf{I})$. When $\bar{\alpha}_T \approx 0$, $\mathbf{x}^T$ is almost Gaussian in distribution, so we have $q(\mathbf{x}^T) \approx \mathcal{N}(\mathbf{x}^T; \mathbf{0}, \mathbf{I})$.

2.2.2 _Denoising Process._ The reverse process aims to recover the original data sample $\mathbf{x^0}$ from the completely noisy $\mathbf{x}^T \sim \mathcal{N}(\mathbf{0}, \mathbf{I})$. The reverse process is formulated as:

$$p_\theta\left(\mathbf{x}^0, \cdots, \mathbf{x}^T\right) = p\left(\mathbf{x}^T\right)\prod_{t=1}^{T} p_\theta\left(\mathbf{x}^{t-1} \mid \mathbf{x}^t\right),$$
$$p_\theta\left(\mathbf{x}^{t-1} \mid \mathbf{x}^t\right) = \mathcal{N}\left(\mathbf{x}^{t-1}; \boldsymbol{\mu}_\theta\left(\mathbf{x}^t, t\right), \Sigma_\theta\left(\mathbf{x}^t, t\right)\right),$$

where $\boldsymbol{\mu}_\theta\left(\mathbf{x}^t, t\right)$ and $\Sigma_\theta\left(\mathbf{x}^t, t\right)$ are the mean and the variance parameterized by $\theta$.

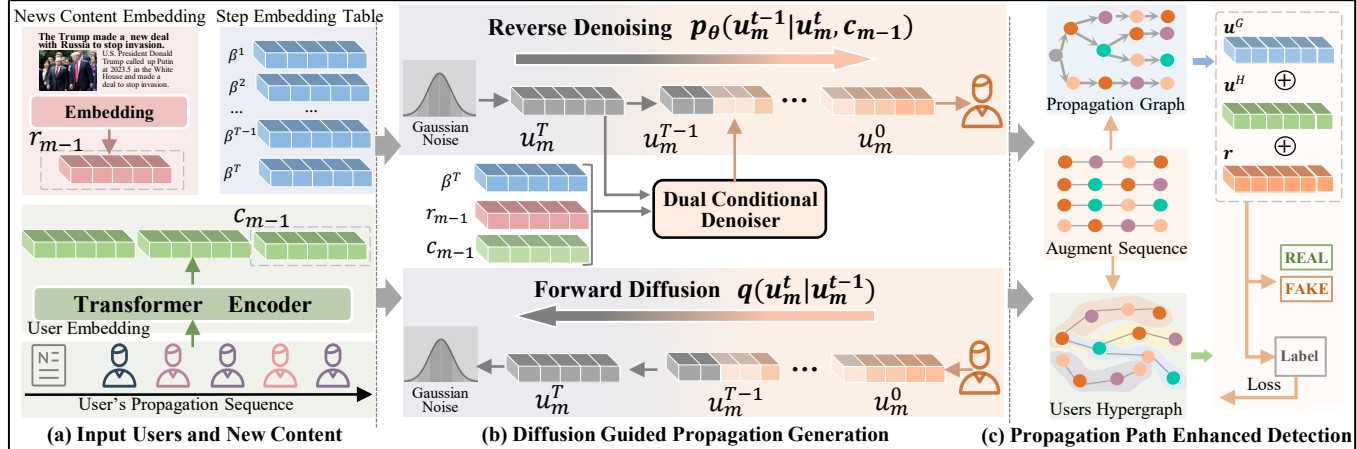

Figure 2: The overview framework of the proposed *DGA-Fake* model.

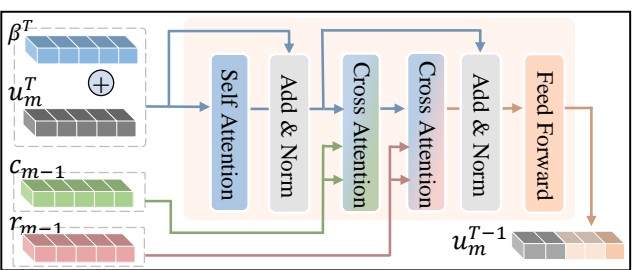

Figure 3: Dual Conditional Denoiser.

## 3 METHODOLOGY

Previous early detection models often prioritize integrating additional social context as a supplement to insufficient propagation paths. However, such approaches are limited by their reliance on time-consuming and labor-intensive auxiliary information, particularly in data-sparse scenarios. In this paper, we propose a method for generating propagation paths in the early stages of news dissemination. To enhance detection capabilities while mitigating social hazards, we introduce a novel approach called Diffusion Guided Propagation Augmentation for Fake News Detection (*DGA-Fake*) in Figure 2. *DGA-Fake* generates propagation sequences based on content features and historical interactions. Following the diffusion guided propagation generation, we further propose a propagation path enhanced detection module (depicted in Figure 4) to integrate propagation global structure and temporal sequence information and detect the fake news at early stage.

## 3.1 Diffusion Guided Propagation Generation

At the early stage of news dissemination, the user interaction is insufficient to provide extra social context, thereby limiting the efficacy of detection performance. Consequently, we propose a propagation path generator to simulate user interaction by a guided diffusion module at data sparse situation. In this section, we first introduce the dual conditional denoiser, a module to integrate control signals into diffusion, as illustrated in Figure 3. Subsequently, we describe the training phase (Section 3.1.2) and generation phase (Section 3.1.3) based on denoising diffusion model.

*3.1.1 Dual Conditional Denoiser.* The vanilla denoising diffusion probabilistic model is not feasible to generate reasonable propagation paths. This is primarily because the generation process lacks controllable conditions from news content and historical sequences. To address this issue, we design a dual conditional denoiser module to integrate the multi-condition features and provide guidance for propagation path generation, spanning from random noise to user embedding.

Firstly, we employ the Transformer encoder to encode the sequential path, $\mathbf{c}_{m-1} = \text{Transformer}(n_0, u_1, \cdots, u_{m-1})$, where the $\mathbf{c}_{m-1}$ denotes the embedding of the historical sequence.

For the news content input, we embed the textual and visual content as $\mathbf{r}_{m-1} = [\mathbf{n}^T || \mathbf{n}^V]$, where $||$ is the concatenation operation. The one-denoising step guided with dual conditions is as follows:

$$
\begin{aligned}
\hat{\mathbf{u}}_m^t &= \text{LN}(\text{SelfAtt}([\mathbf{u}_m^t || \beta^t]) + [\mathbf{u}_m^t || \beta^t]), \\
\hat{\mathbf{c}}_{m-1} &= \text{LN}(\text{CrossAtt}(\hat{\mathbf{c}}_{m-1}, \hat{\mathbf{u}}_m^t) + \hat{\mathbf{u}}_m^t), \\
\hat{\mathbf{r}}_{m-1} &= \text{LN}(\text{CrossAtt}(\mathbf{r}_{m-1}, \hat{\mathbf{c}}_{m-1}) + \hat{\mathbf{c}}_{m-1}), \\
\mathbf{u}_m^{t-1} &= \text{LN}(\text{FF}(\hat{\mathbf{r}}_{m-1}) + \hat{\mathbf{u}}_m^t),
\end{aligned}
$$

where SelfAtt is the self-attention, CrossAtt is the cross-attention, LN is the layer norm and FF is the feed-forward layer. The user embedding $\mathbf{u}_m^t$ and step embedding $\beta^t$ are concatenated firstly. Then historical sequence embedding $\mathbf{r}_{m-1}$ and news content embedding $\mathbf{c}_{m-1}$, are integrated by CrossAtt respectively. The $\mathbf{u}_m^{t-1}$ is the denoising user embedding from step $t$ to step $t-1$. We integrate the dual conditions features into noise embedding through cross-attention operation and control the denoising process to generate reasonable user embedding.

*3.1.2 Conditional Diffusion Training Phase.* In the training phase, rather than applying denoising process from uncontrollable, we reconstruct it with dual conditional denoiser to incorporate multi-conditions. With the guide of news content and sequential path, the denoising process is as follows:

$$
\begin{aligned}
p_\theta(\mathbf{u}_m^{t-1} | \mathbf{u}_m^t, \mathbf{c}_{m-1}, \mathbf{r}_{m-1}) = \\
\mathcal{N}(\mathbf{u}_m^{t-1}; f_\theta(\mathbf{u}_m^t, \mathbf{c}_{m-1}, \mathbf{r}_{m-1}, t), \Sigma_\theta),
\end{aligned} \quad (4)
$$

---

**Algorithm 1:** Guided Diffusion Training

**Input** : News dataset $\{n\}$, users set $\{\mathcal{S}\}$

**output** : Well trained $f_\theta(\mathbf{u}_m^t, \mathbf{c}_{m-1}, t)$

1 *Initialize parameters of $f_\theta$;*

2 **for** *each news $n \in \{n_1, n_2, \cdots, n_N\}$* **do**

    /* Sample and embed the users.     */

3     Target user: $u_m^0, \{n_0, u_1, u_2, \cdots, u_{m-1}\} \sim \mathcal{S}$;

    /* Encode dual conditional signals.     */

4     $\mathbf{c}_{m-1} = \text{Transformer}(n_0, u_1, \cdots, u_{m-1})$;

5     $\mathbf{r}_{m-1} = [\mathbf{n}^T || \mathbf{n}^V]$;

    /* Perform no guided training with probability $\lambda$     */

6     With probability $\lambda$: $f_\theta(\mathbf{u}_m^t, \Phi, t)$;

    /* Sample the diffusion step and generate Gaussian noise.     */

7     $t \sim \text{Uniform}(\{1, 2, \ldots, T\})$,   $\boldsymbol{\epsilon} \sim \mathcal{N}(\mathbf{0}, \mathbf{I})$;

    /* Diffuse the target user with Gaussian noise.     */

8     $\mathbf{u}_m^t = \sqrt{\bar{\alpha}_t}\mathbf{u}_m^0 + \sqrt{1 - \bar{\alpha}_t}\boldsymbol{\epsilon}$;

    /* Take gradient descent step, $\mu$ denotes the step size. */

9     $\theta = \theta - \mu \nabla_\theta \left\| \boldsymbol{\epsilon} - f_\theta(\mathbf{u}_m^t, \mathbf{c}_{m-1}, \mathbf{r}_{m-1}, t) \right\|^2$;

10 **end**

---

where the architecture of $f_\theta(\mathbf{u}_m^t, \mathbf{c}_{m-1}, \mathbf{r}_{m-1}, t)$ is the Dual Conditional Denoiser, $t$ is the time step.

Similar to Eq. 2, the forward diffusion is formulated as a Markov chain of Gaussian transitions:

$$q(\mathbf{u}_m^t | \mathbf{u}_m^{t-1}) = \mathcal{N}(\mathbf{u}_m^t; \sqrt{1 - \beta_t}\mathbf{u}_m^{t-1}, \beta_t \mathbf{I}), \quad (5)$$

where the $[\beta_1, \beta_2, \ldots, \beta_T]$ is the variance schedule.

The objective of propagation path generation is to optimize the variational bound of negative log-likelihood, which equals minimizing the KL-divergence between $q\left(\mathbf{u}_m^{0:T}\right)$ and $p_\theta\left(\mathbf{u}_m^{0:T}\right)$:

$$\mathbb{E}\left[-\log p_\theta(\mathbf{u}_m^0)\right]$$
$$\leq D_{KL}\left(q\left(\mathbf{u}_m^0, \mathbf{u}_m^1, \cdots, \mathbf{u}_m^T\right) \| p_\theta\left(\mathbf{u}_m^0, \mathbf{u}_m^1, \cdots, \mathbf{u}_m^T\right)\right)$$
$$= \sum_{t=1}^{T} D_{KL}\left(q(\mathbf{u}_m^{t-1}|\mathbf{u}_m^t, \mathbf{u}_m^0) \| p_\theta(\mathbf{u}_m^{t-1}|\mathbf{u}_m^t)\right) + C,$$

where $C$ is the constant. For the simplified objective, we randomly sample a fix number of $t$ and minimize the KL-divergence in each iteration, which can be expressed as follows:

$$L_{\text{simple}}(\theta) =$$
$$\mathbb{E}_{\mathbf{u}_m^0, \boldsymbol{\epsilon}} \left[ \left\| \boldsymbol{\epsilon} - f_\theta\left(\sqrt{\bar{\alpha}_t}\mathbf{u}_m^0 + \sqrt{1 - \bar{\alpha}_t}\boldsymbol{\epsilon}, \mathbf{c}_{m-1}, \mathbf{r}_{m-1}, t\right) \right\|^2 \right].$$

More detail about the optimization is displayed in Appendix A.1. The algorithm for conditional diffusion training is presented in Algorithm 1. The procedure starts with encoding the historical sequence $\mathbf{c}_{m-1}$ and news content $\mathbf{r}_{m-1}$ in lines 1-5. Next, the Gaussian noise is generated from the sampled step $t$ and the target user embedding is diffused by noise in lines 6-8. Finally, the parameter $\theta$ in $f_\theta$ is optimized by gradient descent in line 9.

*3.1.3 Propagation Path Generation Phase.* In the generation phase, we target to generate reasonable and informative propagation paths, given the news content and historical sequence. Inspired by [6], we jointly introduce an additional unconditional model using classifier-free guidance scheme [10]. Specially, the guidance conditions $\mathbf{c}_{m-1}$ and $\mathbf{r}_{m-1}$ are replaced randomly by a dummy token $\Phi$ with the probability $\lambda$ to represent as the unconditional diffusion model.

---

**Algorithm 2:** Propagation Generation

**Input** : News $n$ and propagation path $\mathcal{P}$.

**output** : The augment propagation $\mathcal{P}'$.

1 *Number of data augmentation executed $D$;*

2 **for** $d = D, D - 1, \ldots, 1$ **do**

    /* Sample user sequence and noise     */

3     $\{n_0, u_1, \cdots, u_{m-1}\} \sim \mathcal{P}$,   $\mathbf{u}_m^T \sim \mathcal{N}(\mathbf{0}, \mathbf{I})$;

    /* Encode dual conditional signals.     */

4     $\mathbf{c}_{m-1} = \text{Transformer}(n_0, u_1, \cdots, u_{m-1})$;

5     $\mathbf{r}_{m-1} = [\mathbf{n}^T || \mathbf{n}^V]$;

6     **for** $t = T, T - 1, \ldots, 1$ **do**

        /* Control guidance strength.     */

7         $\tilde{f}_\theta(\mathbf{u}_m^t, \mathbf{c}_{n-1}, \mathbf{r}_{n-1}, t) =$
        $(1 + w) f_\theta(\mathbf{u}_m^t, \mathbf{c}_{n-1}, \mathbf{r}_{n-1}, t) - w f_\theta(\mathbf{u}_m^t, \Phi, t)$;

        /* Denoising for one step.     */

8         $\hat{\mathbf{u}}_m^{t-1} = \frac{\sqrt{\bar{\alpha}_{t-1}}\beta_t}{1 - \bar{\alpha}_t}\tilde{f}_\theta(\mathbf{u}_m^t, \mathbf{c}_{n-1}, t) + \frac{\sqrt{\bar{\alpha}_t}(1-\bar{\alpha}_{t-1})}{1-\bar{\alpha}_t}\mathbf{u}_m^t + \sqrt{\tilde{\beta}_t}\mathbf{z}$;

9     **end**

10     Retrieve $\mathbf{u}_m^0$ with Top-$K$ nearest embedding and generate $K$ sequences to update $\mathcal{P}$;

11 **end**

---

To manipulate the influence of the guidance conditions, the $f_\theta$ is modified as the following format:

$$\tilde{f}_\theta(\mathbf{u}_m^t, \mathbf{c}_{m-1}, \mathbf{r}_{m-1}, t) =$$
$$(1 + w) f_\theta(\mathbf{u}_m^t, \mathbf{c}_{m-1}, \mathbf{r}_{m-1}, t) - w f_\theta(\mathbf{u}_m^t, \Phi, t),$$

where $w$ is a hyperparameter to control the influence of conditions. A higher $w$ can enhance conditional guidance, but it could potentially undermine diffusion generalization, consequently deteriorating the quality of the generated user embeddings. The one-denoising step process is as follows:

$$\mathbf{u}_m^{t-1} = \frac{\sqrt{\bar{\alpha}_{t-1}}\beta_t}{1 - \bar{\alpha}_t}\tilde{f}_\theta(\mathbf{u}_m^t, \mathbf{c}_{m-1}, \mathbf{r}_{m-1}, t) +$$
$$\frac{\sqrt{\bar{\alpha}_t}(1 - \bar{\alpha}_{t-1})}{1 - \bar{\alpha}_t}\mathbf{u}_m^t + \sqrt{\tilde{\beta}_t}\mathbf{z}, \quad (6)$$

where $\mathbf{z} \sim \mathcal{N}(\mathbf{0}, \mathbf{I})$. After the propagation generation phase, the user embedding $\mathbf{u}_m^0$ is generated by denoising the Gaussian noise sample $\mathbf{u}_m^T \sim \mathcal{N}(\mathbf{0}, \mathbf{I})$ for $T$ steps.

We further simplify the above conditional diffusion generation process as $\mathbf{u}_m^0 = \text{Diff-GEN}(n, \{n_0, u_1, u_2, \cdots, u_{m-1}\})$, where $\mathbf{u}_m^0$ is the generated next user embedding. To identify the particular user within the user set, we utilize an inner product measurement for retrieving the $K$-nearest users from the candidate set. This process results in the generation of $K$ new sequential paths. Subsequently, the propagation path set $\mathcal{P}$ is updated by incorporating the newly generated sequential paths. And then we randomly sample the candidate user sequence path as $S_{i+1} = \{n_0, u_1, \cdots, u_{|S_{i+1}|}\}$, where $|S_{i+1}|$ denotes the sequence length, and the Diff-GEN$(n, S_{i+1})$ is applied autoregressively to generate the next user embedding $u_{|S_{i+1}|+1}$. This generation process is iteratively repeated, resulting in the refined news propagation path set $\mathcal{P}'$.

The algorithm for propagation generation is in Algorithm 2. The procedure starts with sampling the sequence and encoding conditions in lines 1-5. Next, denoising steps are taken for $T$ times in lines 6-9. Finally, the $\mathbf{u}_m^0$ is retrieved with Top-$K$ user embeddings and the set $\mathcal{P}$ is updated in line 10.

## 3.2 Propagation Path Enhanced Detection

After the augmentation propagation paths are generated by denoising phase, we target to learn the propagation global structure and temporal depth features from the sufficient simulation propagation information. Here, we introduce a propagation path enhanced detection module (depicted in Figure 4), which models the propagation path set $\mathcal{P}'$ into the directed graph and sequence hypergraph to capture the structure and temporal features respectively.

*3.2.1 Learning of Temporal Sequence .* To learn the propagation temporal depth information, a sequence hypergraph is modeled from the generation propagation path. The news propagation path set is denoted as $\mathcal{P}' = \{S_1, S_2, \ldots, S_{|\mathcal{P}'|}\}$. We adopt a hypergraph $H = (U^H, E^H)$ to represent each sequential path $S_i$ as a hyperedge. Formally, each hyperedge $e_i \in E^H$ connects all users that in the same sequence $S_i = \{u_0, u_1, \ldots, u_{|S_i|}\}$. After modeling the sequential path into a hyperedge, we use a hypergraph attention network (Hyper-GAT) with two stage aggregation to learn the temporal depth information [7].

**Node-to-edge Attention.** The hyperedge $\mathbf{e_i}$ is represented by aggregating all connected nodes:

$$\alpha_{ik} = \frac{\exp(\mathbf{a}_1^T \hat{\mathbf{u}}_k)}{\sum_{u_p \in e_j} \exp(\mathbf{a}_1^T \hat{\mathbf{u}}_p)}, \hat{\mathbf{u}}_k = \text{ReLU}(\mathbf{W}_1 \mathbf{u}_k^{l-1}),$$

$$\mathbf{e}_i^l = \sigma\left(\sum_{u_k \in e_i} \alpha_{ik} \mathbf{W}_1 \mathbf{u}_k^{l-1}\right),$$

where $\sigma$ is the activation function, $\mathbf{W}_1$ is trainable weight matrix, $\mathbf{a}_1$ is the weight vector. $\alpha_{ik}$ is the attention coefficient, $l$ is the layer of Hyper-GAT.

**Edge-to-node Attention.** Then we integrate all hyperedges $\mathcal{E}_i$ participated by user $u_i$ to update the user node representation:

$$\beta_{ij} = \frac{\exp(\mathbf{a}_2^T \hat{\mathbf{e}}_j)}{\sum_{e_p \in \mathcal{E}_i} \exp(\mathbf{a}_2^T \hat{\mathbf{e}}_p)}, \hat{\mathbf{e}}_j = \text{ReLU}([\mathbf{W}_2 \mathbf{e}_j^l || \mathbf{W}_3 \mathbf{u}_i^{l-1}]),$$

$$\mathbf{u}_i^l = \sigma\left(\sum_{e_j \in \mathcal{E}_i} \beta_{ij} \mathbf{W}_2 \mathbf{e}_j^l\right),$$

where $\mathbf{a}_2$ is weight vector. $\mathbf{W}_2$ and $\mathbf{W}_3$ are trainable weight matrix.

*3.2.2 Learning of Propagation Structure.* To learn the global propagation structure, a directed propagation graph is constructed from propagation paths, which is denoted as $G = (U^G, E^G)$. Each edge $e_i \in E^G$ connects two users in a single sequence. Graph attention network [37] is applied to learn global structure features:

$$\alpha_{ik} = \frac{\exp(\mathbf{a}_3^T \hat{\mathbf{u}}_k)}{\sum_{k \in \mathcal{N}(i)} \exp(\mathbf{a}_3^T \hat{\mathbf{u}}_k)}, \hat{\mathbf{u}}_k = \text{ReLU}(\mathbf{W}_4 \mathbf{u}_k^{l-1} || \mathbf{W}_4 \mathbf{u}_i^{l-1})$$

$$\mathbf{u}_i^l = \sigma\left(\sum_{u_k \in \mathcal{N}_i} \alpha_{ik} \mathbf{W}_4 \mathbf{u}_k^{l-1}\right),$$

where $\mathbf{a}_3$ is the weight vector and $\mathbf{W}_4$ is the trainable weight matrix.

*3.2.3 Optimization.* To generate the graph level features, we pool the node features from hypergraph and directed graph. Then two types of graph features are concatenated with news content feature

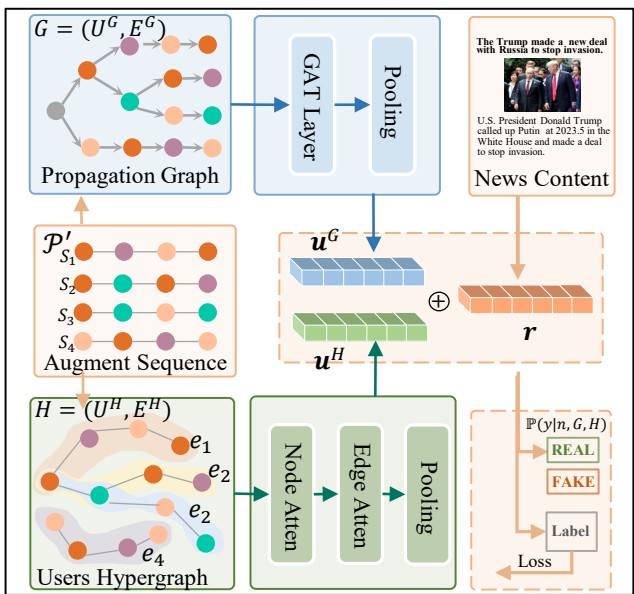

**Figure 4: Propagation Path Enhanced Fake News Detection.**

and fed into a prediction layer to make the final decision:

$$\mathbf{u}^H = \text{readout}(\mathbf{U}^H), \quad \mathbf{u}^G = \text{readout}(\mathbf{U}^G),$$

$$\mathbb{P}(y \mid n, G, \mathcal{H}) = \text{sigmod}(\mathbf{W}_P[\mathbf{u}^H \| \mathbf{u}^G \| \mathbf{r}] + \mathbf{b}),$$

where $\mathbf{u}^H$ is pooled graph level feature as temporal depth feature and $\mathbf{u}^G$ is the global structure feature. The readout is applied by mean pool. Finally, the cross-entropy loss is calculated as:

$$\mathcal{L} = -y \cdot log(\mathbb{P}(y)) - (1-y) \cdot log(1 - \mathbb{P}(y)). \tag{7}$$

## 4 EXPERIMENT

### 4.1 Experimental Settings

**Datasets.** We conduct experiments on three datasets: PolitiFact, GossipCop [30], and Pheme [25], which contain 484, 5,526 and 3,506 news, respectively. The dataset is randomly split into training, validation, and testing sets with a ratio of 7:1:2 [38]. More dataset statistics are presented in the Appendix A.2.

**Baselines.** Two types of SOTA baselines are adopted, including content-based: BERT [5], MVAE [14], EANN [38], CAFE [4], LLAMA2 [36] and Detect-GPT [2]. Propagation-enhanced: CSI [28], Bi-GCN [1], UPFD [8] and MFAN [44]. More detailed descriptions about baselines are listed in the Appendix A.3.

**Implementation Details.** We employ AdamW [20] as the optimizer, and the batch size is set at 16. The hidden size of user embedding is set to 128. More details about the implementation and hyper-parameter are demonstrated in Appendix A.4.

### 4.2 Quantitative Evaluation

Each detection model is executed five times, and the average results are reported in Table 1 and 2. We conduct full propagation paths comparison with propagation-enhanced methods in Table 1. With sufficient propagation information, the propagation-enhanced methods improve detection capabilities. It indicates that propagation paths are capable of providing extra social context to facilitate

| Category | Method | PolitiFact | | | | GossipCop | | | | PHEME | | | |
|---|---|---|---|---|---|---|---|---|---|---|---|---|---|
| | | Acc | Pre | Rec | F1 | Acc | Pre | Rec | F1 | Acc | Pre | Rec | F1 |
| Propagation Enhanced | CSI | 0.753 | 0.834 | 0.742 | 0.758 | 0.782 | 0.823 | 0.783 | 0.808 | 0.771 | 0.825 | 0.831 | 0.828 |
| | Bi-GCN | 0.788 | 0.834 | 0.742 | 0.758 | **0.890** | **0.928** | 0.848 | 0.890 | 0.827 | 0.840 | 0.834 | 0.835 |
| | UPFD | 0.904 | 0.923 | 0.857 | 0.889 | 0.879 | 0.906 | **0.931** | 0.918 | – | – | – | – |
| | MFAN | 0.872 | 0.841 | 0.881 | 0.860 | 0.862 | 0.903 | 0.910 | 0.906 | **0.887** | 0.871 | 0.856 | 0.862 |
| Proposed | DGA-Fake | **0.926** | **0.927** | **0.904** | **0.915** | 0.882 | 0.913 | 0.927 | **0.920** | 0.834 | **0.875** | **0.874** | **0.875** |

Table 1: Performance with full propagation paths. Best results are in bold and second best results are underlined.

| Category | Method | PolitiFact | | | | GossipCop | | | | PHEME | | | |
|---|---|---|---|---|---|---|---|---|---|---|---|---|---|
| | | Acc | Pre | Rec | F1 | Acc | Pre | Rec | F1 | Acc | Pre | Rec | F1 |
| Content Based | BERT | 0.781 | 0.766 | 0.838 | 0.800 | 0.836 | 0.872 | 0.829 | 0.850 | 0.801 | 0.798 | 0.776 | 0.786 |
| | MVAE | 0.812 | 0.803 | 0.835 | 0.819 | 0.782 | 0.802 | 0.751 | 0.776 | 0.776 | 0.734 | 0.722 | 0.728 |
| | EANN | 0.804 | 0.808 | 0.794 | 0.781 | 0.796 | 0.812 | 0.765 | 0.788 | 0.771 | 0.714 | 0.701 | 0.704 |
| | CAFE | 0.848 | 0.857 | 0.850 | 0.853 | 0.832 | 0.804 | 0.903 | 0.851 | – | – | – | – |
| | LLAMA2 | 0.848 | 0.865 | 0.780 | 0.820 | 0.825 | 0.811 | 0.872 | 0.840 | – | – | – | – |
| | Detect-GPT | 0.858 | 0.888 | 0.780 | 0.831 | 0.831 | 0.833 | 0.884 | 0.858 | – | – | – | – |
| Propagation Enhanced | CSI | 0.734 | 0.774 | 0.571 | 0.658 | 0.753 | 0.748 | 0.764 | 0.756 | 0.754 | 0.799 | 0.839 | 0.819 |
| | Bi-GCN | 0.745 | 0.696 | 0.762 | 0.727 | 0.826 | 0.871 | 0.881 | 0.876 | 0.771 | 0.825 | 0.831 | 0.828 |
| | UPFD | 0.862 | 0.892 | 0.786 | 0.835 | 0.822 | 0.860 | 0.891 | 0.875 | – | – | – | – |
| | MFAN | 0.840 | 0.846 | 0.787 | 0.815 | 0.822 | 0.861 | 0.888 | 0.875 | 0.802 | 0.853 | 0.846 | 0.850 |
| Proposed | DGA-Fake | **0.904** | **0.923** | **0.857** | **0.889** | **0.867** | **0.898** | **0.922** | **0.910** | **0.821** | **0.859** | **0.874** | **0.866** |
| | *Imp.(%)* | *+4.2* | *+3.1* | *+0.7* | *+3.6* | *+3.5* | *+2.7* | *+3.1* | *+3.4* | *+1.9* | *+0.6* | *+2.8* | *+1.6* |

Table 2: Performance with early propagation paths. Best results are in bold and second best results are underlined.

| Abaltion Categoty | Method | PolitiFact | | | | GoosipCop | | | | Pheme | | | |
|---|---|---|---|---|---|---|---|---|---|---|---|---|---|
| | | Acc | Pre | Rec | F1 | Acc | Pre | Rec | F1 | Acc | Pre | Rec | F1 |
| Proposed | DGA-Fake | **0.926** | **0.927** | **0.904** | **0.915** | **0.882** | **0.913** | **0.927** | **0.920** | **0.834** | **0.875** | **0.874** | **0.875** |
| Conditional Diffusion | w/o all condition | 0.893 | 0.921 | 0.833 | 0.875 | 0.859 | 0.895 | 0.916 | 0.905 | 0.815 | 0.865 | 0.853 | 0.859 |
| | w/o news content | 0.904 | 0.923 | 0.857 | 0.889 | 0.864 | 0.887 | 0.933 | 0.909 | 0.823 | 0.873 | 0.857 | 0.865 |
| | w/o user sequence | 0.914 | 0.927 | 0.857 | 0.891 | 0.870 | 0.895 | 0.932 | 0.913 | 0.820 | 0.870 | 0.857 | 0.863 |
| Dual Conditional Denoiser | cross-attention | 0.904 | 0.923 | 0.857 | 0.889 | 0.879 | 0.906 | 0.932 | 0.919 | 0.831 | 0.868 | 0.877 | 0.873 |
| | concat+MLP | 0.883 | 0.918 | 0.810 | 0.861 | 0.867 | 0.904 | 0.914 | 0.909 | 0.818 | 0.875 | 0.846 | 0.860 |
| Propagation Path Enhanced Module | w/o directed graph | 0.914 | 0.927 | 0.857 | 0.889 | 0.863 | 0.886 | 0.933 | 0.909 | 0.828 | 0.871 | 0.868 | 0.870 |
| | w/o hypergraph | 0.904 | 0.923 | 0.857 | 0.899 | 0.875 | 0.895 | 0.939 | 0.917 | 0.821 | 0.868 | 0.862 | 0.864 |
| | sequnce encoder | 0.893 | 0.921 | 0.833 | 0.875 | 0.851 | 0.888 | 0.911 | 0.899 | 0.819 | 0.862 | 0.865 | 0.864 |

Table 3: Ablation studies of *DGA-Fake* on three datasets.

the detection of fake news. The *DGA-Fake* achieves the competitive results with propagation-enhanced baselines. Specially, our proposal outperforms all baselines on PolitiFact dataset. It demonstrates that generated propagation paths are capable of providing more context information even in the data sufficient scenarios.

In order to evaluate the detection capabilities of *DGA-Fake* at early propagation stage, we conduct the comparison with early detection baselines in Table 2. To simulate early detection scenarios in the test dataset, we remain the early user engagements in the range of 3 to 5 randomly as the accessible propagation paths. One can clearly see that the performances of propagation-enhanced methods decline a lot and even perform worse than some content-based methods (CAFE). This indicates insufficient propagation paths are difficult to capture the complex global structure and temporal sequence information. Besides, we further compare with content-based large language model baselines LLAMA2 [36] and Detect-GPT [2], which is deployed through the zero-shot prompt engineering. Notably, the LLM-based approaches performs worse than propagation-enhanced methods (in Table 1) and our *DGA-Fake*. It indicates that the textual features alone prove insufficient in adequately verifying the veracity of news. After introducing generated propagation paths by *DGA-Fake*, detection performance is further improved, revealing the pivotal role of sufficient propagation paths. Specially, *DGA-Fake* consistently outperforms all baselines over three datasets, achieving +4.2% on PolitiFact, +3.5% on GossipCop, and +1.9% on Pheme in terms of accuracy. Our proposal generates propagation paths based on news content, causing low social hazard while outperforming propagation-enhanced methods in early stage detection.

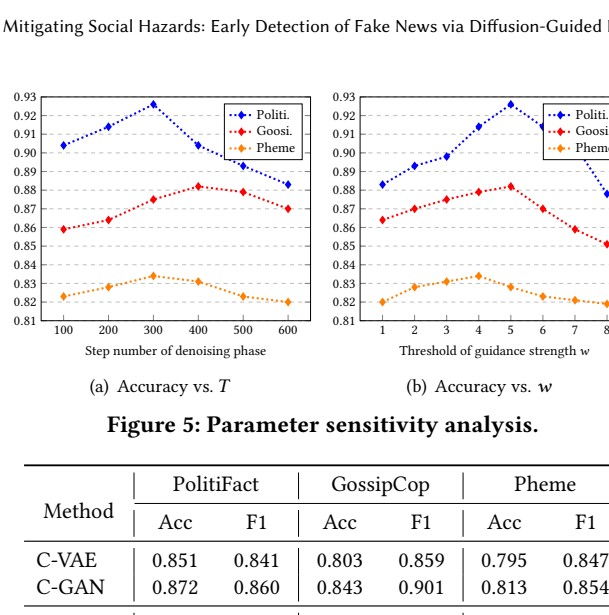

(a) Accuracy vs. $T$       (b) Accuracy vs. $w$

**Figure 5: Parameter sensitivity analysis.**

| Method | PolitiFact | | GossipCop | | Pheme | |
|---|---|---|---|---|---|---|
| | Acc | F1 | Acc | F1 | Acc | F1 |
| C-VAE | 0.851 | 0.841 | 0.803 | 0.859 | 0.795 | 0.847 |
| C-GAN | 0.872 | 0.860 | 0.843 | 0.901 | 0.813 | 0.854 |
| Proposed | **0.926** | **0.915** | **0.882** | **0.920** | **0.834** | **0.875** |

**Table 4: Comparison between generative models.**

## 4.3 Ablation Study

The ablation studies are conducted to investigate the importance of different modules in Table 3.

**Conditional Diffusion**. We aim to study the influence of different conditions. "w/o news content", "w/o user history", and "w/o all conditions" stand for without news content, user sequence, and both. (1) The performance declines more with "w/o news content", revealing that news content is more important than user history. (2) Without all conditions, performance declines significantly, proving both conditions are crucial to improve performance.

**Dual Conditional Denoiser**. Here we focus on investigating the effectiveness of dual conditional denoiser. "cross-attention" and "concat+MLP" denote to replace the conditional transformer-based denoising structure. (1) After replacing with "concat+MLP", model performance presents a significant decline, verifying that simple concatenation operation may not learn two conditional signals effectively. (2) Compared with "cross-attention", our proposal achieves better performance, revealing that multi-stacks of transformer structure are capable of capturing the content condition and user sequence simultaneously.

**Propagation Path Enhanced Detection**. "w/o directed graph", "w/o hypergraph" are implemented without directed propagation graph and user hypergraph. The "sequence encoder" is denoted as to replace with user sequence encoder. (1) Both modules are crucial to achieving desirable detection performance as "w/o hypergraph" and "w/o directed graph" only achieve degraded results. It demonstrates both global structure feature and sequence depth feature are beneficial for detection performances. (2) Without the both two graph modules, performance declines significantly due to sequence information can not reflect the propagation temporal and structure information effectively.

## 4.4 Parameter Sensitivity Analysis

Here we conduct hyper-parameter sensitivity analysis on two parameters: the number of diffusion step $T$ and the guidance strength

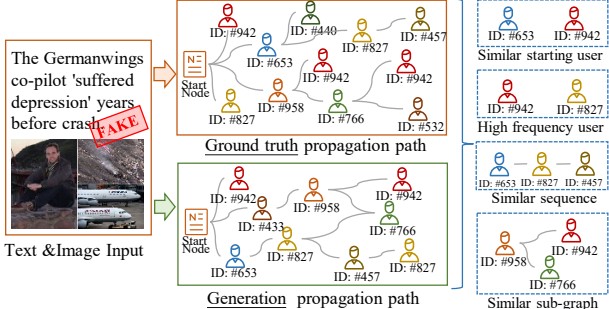

**Figure 6: Case studies on Pheme dataset.**

$w$. In Figure 5(a), with the increase of $T$, model performance first increases and then significantly drops. A greater number of diffusion steps are required to obtain more refinement, while the generated embeddings do not fit the distribution of real propagation paths if $T$ is too small. Figure 5(b) shows that with the increase of conditional guidance strength $w$, model performance first improves and then drops. This demonstrates that too much conditional signal may hurt the quality of the user embedding.

## 4.5 Generative Model Analysis

In this section, different generative models are compared in Table 4. We take the deep generative model C-VAE [23] and C-GAN [17] as baselines. The proposed *DGA-Fake* outperforms the generative models over three datasets. Due to the diffusion-guided model generating diverse and high-quality propagation paths, our proposal is capable of capturing the temporal and structural features to facilitate the detection.

## 4.6 Case Study

Figure 6 presents the case of generated propagation path. One can clearly see that the generated paths are similar to the ground-truth in terms of high-frequency user, initial spreading user, propagation sequence and sub-graph structure. It demonstrates that our proposal learns the real propagation feature in the user community, capturing the temporal and structural features during news spreading.

## 5 RELATED WORK

We divide detection methods into two categories. **Content-based**: many methods focus on content features for detection [13, 21, 22, 33, 39]. Some recently works deploy large language model by prompt engineering [2, 36]. **Propagation-enhanced**: some studies apply propagation path to detect fake news [8, 27, 31, 44]. Besides, some researchers integrate auxiliary information for early detection [19, 24, 32, 42]. More details about related work are in Appendix A.5.

## 6 CONCLUSION

In this paper, we propose a novel Diffusion Guided Propagation Augmentation Fake News Detection model. We generate the propagation paths via conditional diffusion module and further incorporate them with news content feature for fake new detection. Experimental results evaluated on three popular datasets demonstrate the superiority of our proposal.

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
