# OpenReview forum: "Mitigating Social Hazards: Early Detection of Fake News via Diffusion-Guided Propagation Path Generation"
_acmmm.org/ACMMM/2024/Conference — MM2024 Poster_

### Official Review · Reviewer_14oS · 2024-05-13

**Rating:** 3
**Confidence:** 3

**Summary:**

The paper investigates the problem of early fake news detection through propagation path generation, and the authors propose a novel generative model, DGA-Fake, which integrates a Gaussian-based bootstrap diffusion module that helps to progressively denoise and generate plausible user embeddings of the next interactions from randomly sampled Gaussian noise, which can be used to generate a simulated sequence of user interactions guided by the history of the interactions and news content. The authors use the propagation path enhancement detection module to learn temporal depth information through user sequence hypergraphs.DGA achieves early detection of fake news by integrating the propagation of global structure and temporal sequence information.

**Strengths:**

1. The paper introduces a novel approach, DGA-Fake, for early fake news detection by generating propagation paths based on content features and historical interactions. The incorporation of a guided diffusion module to simulate user interactions at the early stage of news dissemination showcases a unique theoretical approach to enhancing detection capabilities while mitigating social hazards.
2.  Strong experiment performance. The technical correctness is supported by the thorough evaluation conducted across three datasets, demonstrating the superiority of the proposed approach.
3. Well-structured paper. The clarity of the paper in explaining the methodology, experimental setup, and results enhances its readability and understanding.

**Limitations:**

1. Insufficient Comparison with State-of-the-Art Methods: The authors did not compare articles on fake news detection after 2023.

2. Lack of clarity on the advantages of diffusion modeling: the authors do not clearly state why they chose to use diffusion modeling for generating user embeddings and paths, and the advantages of diffusion modeling over other approaches.

3. The MM official instructions state that the body section of the article can be 8 pages long, but the author only wrote 7 pages of content, so why not put the content from the supplemental material in the body of the paper?

**Suitability:**

3

---

### Official Review · Reviewer_mEF2 · 2024-05-21

**Rating:** 4
**Confidence:** 3

**Summary:**

This paper proposes enhancing early fake news detection by using propagation path generation for the first time. The authors introduce the DGA-Fake model, which creatively generates reasonable propagation paths through controllable guided diffusion. The model integrates global propagation structure and temporal depth information via those propagation paths to improve fake news detection.

**Strengths:**

1. **Well-Written Manuscript:** The authors clearly explain why using propagation path generation for fake news detection is beneficial. They outline the challenges of this approach and their solutions, making it easy for readers to understand the paper's motivation and main contributions.
2. **Novelty:** By leveraging the innovative approach of using diffusion models to generate propagation paths, the authors offer a new perspective on enhancing the detection process.
3. **Extensive Experiments:** The authors conducted extensive experiments to validate their model’s performance. Overall, it is a solid piece of work.
4. **Code Availability:** The authors provide their code, ensuring that other researchers can replicate and validate the experiments. This transparency is crucial for advancing academic research.

**Limitations:**

1. **Method Details:** The process for obtaining user embeddings is not clarified. Additionally, some variables in the paper lack direct interpretation.
2. **Typos:** There are typographical errors in the paper, such as the abbreviation of "Gossip" in Figure 5 ("Goosi."), "sigmod" in line 548, and the inconsistency of "PHEME" and "Pheme" in Tables 1, 2, and 3. Correcting these errors will improve the manuscript's clarity.
3. **Ablation Study:** The authors could provide results for detection performance using only content and only propagation. This would offer a clearer understanding of the enhancement brought by propagation path augmentation.
4. **Dataset Details:** A more detailed description of the datasets would be beneficial. Information such as the average number of propagation sequences per news post and the average length of these sequences would help readers better understand the data condition.
5.  **Paper Structure**: The authors have compressed the main content to 7 pages ( the page limit is 8 in fact), placing many details in the appendix. Some of this supplementary content could be integrated into the main text to enhance the paper’s readability and coherence.

**Suitability:**

2

---

### Official Review · Reviewer_rr1C · 2024-05-25

**Rating:** 4
**Confidence:** 3

**Summary:**

The authors investigated the early detection of fake news by simulating realistic propagation paths based on news content before
actual spreading. Specifically, they proposed a generative model, DGA-Fake, which generates the propagation paths via a conditional diffusion module and further incorporates them with news content features for fake news detection. Evaluation across three datasets demonstrates the effectiveness of their proposed model.

**Strengths:**

This paper is well-motivated and provides new ideas for early rumor detection by predicting propagation paths.
This paper is well organized, with algorithms and formulas clearly detailing the flow and structure of the proposed model.

**Limitations:**

The methodology section involves numerous hyperparameters, but the parameter analysis part is somewhat insufficient. And since you're utilizing hypergraphs in your method, it might be more comprehensive to include a hypergraph-based method to Propagation-enhanced methods as baselines.
There are some typos. In Figure 4, it should be e3 instead of e2, And “w/o all conditions” stands for without news content, user sequence, and both.” In 4.3 seems repetitive.

**Suitability:**

2

---

### Official Review · Reviewer_s8iV · 2024-05-30

**Rating:** 3
**Confidence:** 3

**Summary:**

This paper proposes a novel approach called DGA-Fake for early detection of fake news on social media platforms using diffusion-guided propagation path generation. Unlike previous methods that rely solely on content analysis or require extensive propagation data, DGA-Fake generates realistic propagation paths based on news content and historical user interactions in the early stages of news dissemination. The model employs a dual conditional denoiser module to guide the generation process and incorporates the generated paths into a propagation-enhanced detection module to capture global structure and temporal features. Experiments on three real-world datasets demonstrate the superiority of DGA-Fake compared to state-of-the-art baselines.

**Strengths:**

Pros:
1. Addresses the dilemma between detection efficacy and social hazard by generating propagation paths in the early stages, reducing the potential harm caused by the spread of fake news.
2. Utilizes a novel diffusion-guided approach to generate realistic propagation paths based on news content and historical user interactions, providing rich social context for detection.
3. Incorporates a dual conditional denoiser module to effectively control the generation process and integrate multi-condition features.
4. Employs a propagation path enhanced detection module to capture global structure and temporal depth information from the generated paths, improving detection performance.
5. Consistently outperforms state-of-the-art baselines on three real-world datasets, demonstrating its effectiveness in early fake news detection.

**Limitations:**

Cons:
1. The main concern is that the related works are highly insufficient. Highly related references [1,2,3,4] are missing. It is strongly suggested the authors may comprehensively discuss the related works in the related work section, even in the appendix. Otherwise, this paper may mislead the readers.
2. It is not clear whether or not the authors conduct multiple runs for each results. Standard deviation is desired if the author conduct multiple runs. Otherwise, the results are not convincing.
3. Statistical testing is desired. It is highly recommended that the authors conduct a Statistical testing. Otherwise, the statistical significance of the results are not clear.
4. It is unclear whether the generated Propagation Path actually matches the real-world Propagation Path. More explanations in this aspect are desired. It is also suggested that the authors may verify the accuracy of generated Propagation Path. Otherwise, the motivation is not convincing enough.

[1] A Multitask, Multi-lingual, Multimodal Evaluation of ChatGPT on Reasoning, Hallucination, and Interactivity. https://arxiv.org/abs/2302.04023

[2] Towards Reliable Misinformation Mitigation: Generalization, Uncertainty, and GPT-4 EMNLP 2023 https://arxiv.org/abs/2305.14928

[3] Fact-Checking Complex Claims with Program-Guided Reasoning ACL 2023 https://aclanthology.org/2023.acl-long.386/

[4] Combating Misinformation in the Age of LLMs: Opportunities and Challenges https://arxiv.org/abs/2311.05656

**Suitability:**

2

---

### Meta-Review · Area_Chair_tsvu · 2024-07-04

**Recommendation:** Accept (Poster)
**Confidence:** 3

**Metareview:**

This paper focuses on multi-modal fake news detection leveraging not only the content but also the propagation network structures. To address the propagation structure incompleteness at the early stage of fake news spread, the authors propose to generate the propagation paths based on the guidance of the diffusion process. All reviewers acknowledge the contribution regarding the idea, the method design, and good experimental results. The overall writing is of good quality. However, they also expressed their concerns regarding the result significance, the sufficiency of related work citing and discussing, the sufficiency of experiments, and the paper structure (only 7 pages for main content considering an 8-page main content limitation).

After rebuttal, no reviewers chose to change their recommendation. The two reviewers providing negative ratings still have concerns about the recency of the added baselines and the revision workload to make it to be a full 8-page paper when preparing the next version. Overall, I believe this submission provides a novel idea and has a good execution of method design and experiments after rebuttal. By checking the updated version in the anonymous GitHub repository, the workload risk of revision is finally lowered. I did not find very strong and critical aspects to reject this paper if every revisions and additions are well presented in the final version.